# Characteristics and Management of Children with Appendiceal Neuroendocrine Neoplasms: A Single-Center Study

**DOI:** 10.3390/cancers16203440

**Published:** 2024-10-10

**Authors:** Stefano Mastrangelo, Giorgio Attinà, Guido Rindi, Alberto Romano, Palma Maurizi, Antonio Ruggiero

**Affiliations:** 1Pediatric Oncology Unit, Department of Woman and Child Health Sciences and Public Health, Fondazione Policlinico Universitario A. Gemelli-IRCCS, 00168 Rome, Italy; giorgio.attina@policlinicogemelli.it (G.A.); alberto.romano@guest.policlinicogemelli.it (A.R.); palma.maurizi@unicatt.it (P.M.); antonio.ruggiero@unicatt.it (A.R.); 2Department of Life Sciences and Public Health, Università Cattolica del Sacro Cuore, 00168 Rome, Italy; guido.rindi@unicatt.it; 3Anatomic Pathology Unit, Department of Woman and Child Health Sciences and Public Health, Fondazione Policlinico Universitario A. Gemelli-IRCCS, 00168 Rome, Italy

**Keywords:** appendiceal tumor, neuroendocrine tumor, rare tumors, children, appendectomy, pediatric oncology

## Abstract

**Simple Summary:**

Appendiceal neuroendocrine neoplasms are very rare in pediatric populations; thus, there are no common guidelines or consensus for the management in children and adolescents, and adult protocols are usually adopted. In the present study, we report on 17 patients that underwent appendectomy for appendicitis with incidental diagnosis of a neuroendocrine tumor revealed upon histologic examination. Patients’ characteristics, tumor histology, follow-up and outcome are described and compared to data from the literature, with similar results being reported in various studies. Analysis of previous scientific publications on children with appendiceal neuroendocrine neoplasms was performed and recommendations for treatment and follow-up were reviewed. This tumor displays benign behavior and an excellent outcome in children and adolescents; thus, many authors agree that aggressive surgery after the patient’s first appendectomy is not necessary and follow-ups can be reduced.

**Abstract:**

Background/Objectives: Appendiceal neuroendocrine neoplasms (ANENs) are usually found incidentally during histology examination after appendectomy for appendicitis. Due to their rarity in pediatric populations, there is no consensus on treatment or follow-up. The analysis of patients with ANENs of our and other studies will increase the understanding of this tumor. Methods: Pediatric patients with ANENs were uniformly managed at our center between 1998 and 2023. Patients’ presenting symptoms, surgery, tumor histology, post-surgical work-up, follow-up and outcome were analyzed. Results: Our report describes 17 patients with a diagnosis of ANEN after appendectomy. The median age was 14 years (range of 4–17 years). Tumors were located at the tip of the appendix in 58.8% of cases and only one had a diameter >1 cm. All were well-differentiated tumors with free resection margins. The submucosa was invaded in five cases, muscularis propria in eight and subserosa in four. Post-appendectomy work-up included tumor marker measurement, abdominal ultrasound and computed tomography or magnetic resonance imaging, chest X-ray and octreotide scintigraphy. No residual tumors or metastases were detected. Additional surgery was not necessary. Follow-up was carried out for a median duration of 6 years (range of 1–10 years). Only one patient was lost to follow-up and all other patients are alive without tumor recurrence. Conclusions: The tumor characteristics of our patients confirmed data from the literature. With the lack of a sufficient number of large prospective trials, it is important to add more information to confirm the benign nature and excellent outcome of this tumor, even without additional surgery. Consensus guidelines are needed for ANENs in pediatric populations.

## 1. Introduction

Neuroendocrine neoplasms are a heterogeneous group of tumors that can present in different organs of the body, including the gastrointestinal tract, pancreas, lungs and other rarer locations such as the thymus and adrenal glands. They arise from neuroendocrine cells found in the gastrointestinal tract and tracheobronchial tree. These neoplasms were first described in 1867 by Lubarsch, a Swiss pathologist, who observed multiple tumors in the distal ileum of two patients during an autopsy [1].

In 2010, the World Health Organization classified NENs based on their differentiation and proliferative grading, with epithelial well-differentiated neoplasms defined as neuroendocrine tumors and epithelial poorly differentiated neoplasms defined as neuroendocrine carcinomas, and mixed adenoneuroendocrine carcinomas [2].

The clinical presentation of neuroendocrine tumors depends on their ability to secrete serotonin and other neuropeptides, which determines their classification as functional or non-functional tumors. Functional tumors exhibit specific signs and symptoms related to the substances they produce, such as histamine, bradykinin, serotonin, and 5-hydrotryptophan causing carcinoid syndrome. These substances are metabolized by the liver; as a consequence, carcinoid syndrome manifests only in the presence of liver metastasis [3,4,5]. In contrast, non-functional tumors usually remain asymptomatic and are diagnosed incidentally or, if large, after causing local pain at the primary site [5].

In the global adult population, the appendiceal neuroendocrine neoplasm (ANEN) incidence was found to range from 9.7 to 14/million/year [6]. Small tumors are considered to have benign behavior; nevertheless, in cases of patients with tumors 2 cm in size or greater, the 5-year mortality is 29.5%, and for 1 cm tumors, the rate is 5% [7].

On the contrary, ANENs are extremely uncommon in the global pediatric population and have benign behavior. Their incidence is 1 to 1.14/million/year, and they represent 0.1% of all pediatric malignancies [8]. Nevertheless, ANENs are the most common gastrointestinal tumors in children and adolescents [9].

In children with appendicitis, after surgical removal, ANENs are found incidentally during histologic examination; their prevalence ranges from 0.09% to 1.5% of appendectomies [10]. It is very difficult to preoperatively diagnose tumors as imaging performed for appendicitis most likely will not disclose the presence of these usually very small tumors [11,12]. As an exception, Zeineddin et al. [13] reported one case that was identified during preoperative imaging for a total of 147 patients with ANENs.

A normal appendiceal wall is formed of four layers, the serosa, subserosal fat tissue, muscularis propria, submucosa and mucosa. Neuroendocrine cells are located in the mucosa (epithelial neuroendocrine cells) and in the submucosa (subepithelial neuroendocrine cells).

Neuroendocrine tumors can arise from any part of the appendiceal wall and be located from the base to the tip of the appendix [4]. Macroscopically, they appear as solid, greyish-white masses that become yellowish after fixation, and are typically well defined, but lack a true capsule. Their diameter is less than 1 cm in 48% of cases, whereas large tumors (>2 cm) are uncommon, representing only 4% of the total [14].

After diagnosis, all patients should be evaluated for possible residual tumor mass or metastasis by the determination of specific markers and imaging [15].

Chromogranin A (CgA) is a 49 kd acidic polypeptide released by neuroendocrine cells present in the body [7]. Its normal range is 27 to 94 ng/mL. The serum CgA level may be associated with tumor burden, with smaller (<2 cm) ANENs often having normal CgA levels. Due to its high specificity, it can be used as a tumor marker for ANENs [7]; however, the role of CgA in monitoring disease recurrence has not yet been established [16,17,18]. 5-Hydroxyindoleacetic acid (5-HIAA) is the main metabolite of serotonin. It is tested during urine collection and a normal range is 0.5 to 10 mg/24 h. It can replace serotonine measurement in cases of carcinoid syndrome, since it does not vary with patient’s stress and activity [7]. Nevertheless, because of the low sensitivity of 5-HIAA, it should not be used as a screening test, but only to monitor possible recurrence of ANENs during follow-up in patients with a documented diagnosis, even though its use is still controversial [9].

As it is uncommon for patients to undergo specific pre-operative imaging investigations, imaging has a minimal role in diagnosis, but it is used in staging to search for residual tumors or to rule out the presence of metastases, and in follow-up for possible tumor recurrence. The most used methods are abdominal ultrasound (US), abdominal computed tomography (CT), and magnetic resonance imaging (MRI) [7,18].

Furthermore, as neuroendocrine neoplasms express somatostatin receptors, the most specific imaging modality for detecting ANEN lesions is Octreonscan, a type of octreotide scintigraphy, using indium-111-labeled somatostatin analogues [15]. Nevertheless, its use has not demonstrated to be useful in identifying small ANENs of less than 1 cm [19,20]. Currently, Octreoscan is recommended for patients with known ANENs and elevation of urinary 5-HIAA or a suspicious mass on US [15]. It is important to consider that Octreoscan can yield false-positive results due to respiratory infections, concurrent granulomatous disease, adrenal uptake, accessory spleen, and surgical scars [21]. A more reliable method to detect neuroendocrine tumors seems to be Ga-68-DOTATATE PET/CT [22], although its use has not been reported in pediatric ANEN papers.

Treatment of ANENs in children and adolescents is surgical, although the type and timing of surgery are still a matter of debate, due to the rarity of the neoplasm and the limited number of studies available. Historically, treatment recommendations were developed looking at the experience of adults, although ANENs show different behavior in pediatric populations. The consensus guidelines for adult populations were established by the European Neuroendocrine Tumor Society (ENETS) and published in a 2016 paper [18], which were then updated in 2023 [6], and by the North American Neuroendocrine Tumor Society (NANETS) [7]. Both societies recommend right hemicolectomy (RHC) for patients with high-risk ANENs [6,7,18]. Nevertheless, considering that for unknown reasons, children present a better prognosis, in a paper published by the ENETS, it was stated that adult guidelines do not apply to the pediatric population [18].

After initial appendectomy and diagnostic work-up, there is a lack of consensus in the scientific literature even regarding the optimal follow-up modality and duration for children with ANENs [9], and most advice is derived from adult protocols [7,18,23].

In general, follow-up may include physical examinations, laboratory tests (mainly the tumor markers CgA and 5-HIAA) and imaging studies with abdominal US, CT or MRI, and, if clinically indicated, Octreoscan [15]. Regarding the interval of assessment and duration of follow-up in adults, investigations are suggested every 6 to 12 months up to at least 7 years, due to the indolent nature of this tumor [7,18].

The rationale behind performing follow-up might be the perceived risk of lymph node metastasis development, but, since there is no conclusive evidence to support its effectiveness in preventing tumor recurrence or impacting long-term outcomes, follow-up is not recommended by all authors [11,14].

As pediatric ANENs represent such a rare tumor, it has been difficult to produce comprehensive and standardized guidelines on their treatment and follow-up. Moreover, since the diagnosis is mostly incidental, they are probably underdiagnosed; consequently, it is even more complicated to redact a global study on their behavior and progress [11].

The aims of this paper were to analyze data regarding the patients treated and followed at our hospital, and to identify the most suitable diagnostic, therapeutic and follow-up guidelines for ANENs in pediatric populations in relation to recent literature data.

## 2. Materials and Methods

In this study, we analyzed the characteristics of patients affected by ANENs followed in the Pediatric Oncology Unit at Fondazione Policlinico Universitario A. Gemelli-IRCCS. The study was approved by the Pediatric Institutional Review Board of Università Cattolica del Sacro Cuore, Fondazione Policlinico Universitario A. Gemelli-IRCCS (protocol code: DIPUSVSP-22-09-2325).

Patients under 18 years of age with a histological diagnosis of ANEN were included. After appendectomy, evaluations of urinary 5-HIAA and serum CgA, abdominal US and CT scan, chest X-ray, and Octreoscan were planned for all patients.

According to ENTS and NANTS recommendations (Table 1) and indications suggested in papers regarding pediatric studies, second surgery was not considered for well-differentiated and small tumors (<2 cm), with negative tumor margins, regardless the position or invasion depth in the appendiceal wall.

Our standard follow-up included clinical examination, urinary 5-HIAA and serum CgA dosing, abdominal US and CT scan. Octreoscan was reserved for masses suspected during the other investigations.

The follow-up assessment included intervals of 3 months in the first year and then every 6 months for up to 3 years and once a year thereafter. Follow-up was planned to continue for 10 years.

Information on each patient was collected from the medical database at initial presentation and at follow-up. Analyzed data included patients’ age at diagnosis, gender, initial complaints and symptoms, surgical procedure, histopathological findings, laboratory and imaging results at diagnosis and follow-up, and outcomes.

## 3. Results

We reviewed 17 pediatric patients who underwent appendectomy between 1998 and 2023, with a diagnosis of ANENs. The patients’ characteristics are described in Table 2.

Six patients included in this series were referred to our center after appendectomy and histologic diagnosis performed elsewhere. All histological specimens were reviewed at our institute of Pathology and diagnosis of ANEN was confirmed.

Our group of patients is almost homogeneous in terms of gender (nine males and eight females), with a median age of 14 years (range of 4–17 years). All patients presented definite symptoms of acute appendicitis, with abdominal pain in all cases, while seven patients had fever and four vomited.

By microscopic evaluation, it was found that all were well-differentiated grade 1 (G1) tumors, with a Ki67 index < 2% in the 10 analyzed specimens. Twelve patients had tumors localized at the apex, four in the body and one at the base. The maximal infiltration observed involved the subserosa, which was reported in four cases, whereas the other eight tumors invaded the muscularis propria and five the submucosa. No tumor was extended to the mesoappendix. Tumor size ranged between 0.2 and 1.1 cm, with an average of 0.6 cm. Excision margins were free of tumors in all cases.

All patients underwent post-surgical work-up as planned. The patients did not show any alteration in the biochemical and hematological tests performed, except for patient n. 2 and n. 16 that demonstrated a transient increase in urinary 5-HIAA 1 month (maximum level was 14.3 mg/24 h) and 20 days (maximum level was 18.3 mg/24 h) after surgery, respectively. The value normalized after a few weeks in both patients who did not show any other anomaly during the rest of the follow-up. Two patients (patient n. 4 and n. 6) had evidence of a mass on abdominal US, and were further investigated with US and MRI that showed in both cases the presence of a lymphocele related to appendectomy, which then spontaneously resolved. Due to a previous diagnosis of celiac disease, patient n. 5 underwent colonoscopy with multiple biopsies that were negative for neoplasms.

No residual tumor or metastasis were revealed by abdominal US, chest X-ray, abdominal CT scan or Octreoscan. On the basis of tumor histological characteristics and staging, we decided to avoid additional surgery after the primary appendectomy in all of our patients.

All follow-up imaging and laboratory tests performed in our patients were normal. Octreoscan was never used during follow-up.

Six patients are still undergoing follow-up. After 3 years, three patients expressed the desire to no longer attend follow-up at our hospital, but kept in touch and were contacted for this study. Instead, one patient was lost to follow-up after 1 year from diagnosis and has not been in contact with us since. The other sixteen patients, six of whom are still undergoing follow-up, are currently alive and in complete remission. The mean follow-up time is 6 years, ranging from 1 to 10 years.

## 4. Discussion

In our series, the age of patients was similar to that reported in the literature [14,24]. In total, 53% of patients were male, a percentage slightly different from the 41% rate reported by Van Amstel et al. [25].

All children in our report presented with clinical symptoms of acute appendicitis without carcinoid syndrome as expected, since all displayed localized tumors. This kind of presentation, as reported in other studies, makes the preoperative diagnosis of ANEN very difficult even if abdominal US is performed when acute appendicitis is suspected [26,27].

Furthermore, in recent years, there has been increasing interest in the use of antibiotics for conservative treatment of acute appendicitis, with reported success rates ranging from 62% to 95.5% [28,29]. Moreover, if appendectomy is not addressed in all cases of acute appendicitis, there is a risk of delaying the diagnosis of ANEN [26], as demonstrated in previously reported cases [13,30]. This may lead to the development of advanced disease and the need for non-surgical treatment. In adults, systemic treatment includes somatostatin analogs such as octreotide [31], targeted therapies based on the mTOR inhibitor everolimus [32] and second-line therapy [33].

In the present study, following histologic examination, it was found that the tip of the appendix was the most frequent tumor site (58.8%), whereas the base was reported for only one patient (5.8%). These percentages are comparable to other studies with a larger number of children with ANENs that presented rates of apical, body and base positions of 55.6%, 18.4% and 6.3%, respectively [25]. Tumor grading revealed well-differentiated tumor cells (G1) in all of our patients and low percentages of high-grade tumors (G2–G3) in children were also reported in a similar study [15].

Patients in our study were found to have appendiceal wall infiltration to the submucosa in 29.4% of cases, to the muscularis propria in 47%, and to the subserosa in the remaining patients, although no infiltration of the mesoappendix was observed. In the review by Van Amstel et al. [25], mesoappendix involvement was reported in 27.8% of cases, whereas the submucosa, muscularis propria, and subserosa were invaded by tumors in 26.4%, 18.9%, and 17.8% of cases, respectively. Regarding tumor size, our patients presented a tumor < 1 cm in 16 out of 17 cases, in accordance with other reports in which tumors > 2 cm were rare [24,25]. None of the patients had positive tumor margins following appendectomy, which is similar to the literature rate of 2.5% to 3.1% [14,24].

As soon as the diagnosis of ANENs is confirmed, it is important to perform a thorough diagnostic investigation to look for possible metastasis or residual tumors. In adults, imaging evaluation with abdominal US, abdominal CT, chest X-ray, and Octreoscan is usually used to reveal the disease status [6,7,18].Furthermore, specific markers tests should accompany imaging during post-appendectomy work-up. To reduce the rate of false-negative tests, since urinary 5-HIAA has a specificity of around 88%, but a sensitivity less than 50%, it should be tested together with serum CgA, which has a 70% sensitivity rate [15].

Two of our patients displayed an increase in urinary 5-HIAA levels, which returned to normal in subsequent testing. In addition, the patients’ concomitant CgA levels were always in the normal range; this confirms the low reliability of evaluation using these markers and the need to repeat them.

Another two patients, due to the presence of a suspicious mass on abdominal US, experienced a period of stress with their patients, until a subsequent MRI showed that it was a lymphocele in both cases. This supports the idea suggested by de Lambert et al. [11] that imaging is of little utility and can also trigger further unnecessary and stressful investigations for a tumor that is considered at low risk of recurrence.

Abdominal US is undoubtably useful to detect liver metastasis, but, even in this case, its utility has been questioned since distant ANEN metastasis has never been reported in children [11,14,34]. Furthermore, it has been reported that CT scans and MRI have limited use in the search of ANEN micrometastases in the lymph nodes [18]. Otreoscan has proven to be a valuable tool in defining metastatic disease in children [35]. However, it should be noted that its sensitivity is reduced in lesions smaller than 1 cm [9] and that false-positive results are possible in up to 10% of cases [15].

Nevertheless, in many studies regarding pediatric populations, work-up after diagnosis usually consists of 5-HIAA and CgA evaluation, abdominal US and CT scans, which are less frequently supported by MRI and Octreoscan [14,25].

In a paper by Boxberger et al. [4], about half of the patients underwent urinary 5-HIAA and/or serum CgA measurements; however, it was found that the tests were positive in patients with no tumors following the second surgery and negative in patients with lymph node infiltration. The authors concluded that the tests were not sensitive enough to pick up micrometastases. Octreoscan was adopted in 53 out of a total of 237 patients, but even this method was not considered sufficiently sensitive for routine use.

De Lambert et al. [11] performed post-surgery investigations in 40% of their 114 patients, mostly with abdominal US and CT scans. Only 8 patients underwent an Octreoscan, while 16 had urinary 5-HIAA measurements and 1 had CgA measurements taken, although all were in the normal range.

In an Italian trial [15], all patients were investigated using chest X-ray, abdominal US, Octreoscan, urinary 5-HIAA and serum CgA measurements. CT scans and MRI were reserved for suspicious cases. These investigations were fundamental for making treatment decisions together with the information regarding the completeness of resection and tumor size. The authors concluded that 5-HIAA and CgA should be dosed concomitantly to increase sensitivity. In their trial, 81 Octreoscans were performed (76 in patients with tumors < 2 cm), and only 7 were abnormal. For five of these patients, the following Octreoscan was negative and in the sixth patient, the second surgery did not reveal a tumor. The authors stated that Octreoscan seemed to have a limited role in the work-up and follow-up of ANENs especially with tumors < 1 cm in size, but it may be useful if the initial tumor is >2 cm or if it bears other high-risk factors.

Sommer et al. [9] in their “Swiss algorithm” also rely on post-surgical work-up to decide if RHC is necessary or not.

In their 2016 review of 958 pediatric ANEN cases, Njere et al. [14] counted 377 patients that underwent abdominal US, 371 that underwent urinary 5-HIAA dosing, 209 that underwent CgA tests and 209 that underwent an Octreoscan. CT scans, MRI and chest X-rays were performed in less than 90 cases. The authors concluded that investigations after appendectomy were not useful to decide if further treatment was necessary or not.

Appendiceal neuroendocrine neoplasms are extremely uncommon in pediatric populations; therefore, there are currently no standardized therapeutic indications. For this condition, algorithms have been derived from adult experience to aid the decision-making in children. Nevertheless, not all centers agree in following adult protocols for their pediatric patients, as they appear to be too invasive and aggressive [11,15,23,34]. Moreover, some authors argued that recommendations for second surgery are too aggressive even in adults. It has been stated that in the global adult population, RHC does not provide significant survival benefits even if tumors have a diameter > 2 cm, as appendectomy alone is sufficient for local disease control [36,37]. Also, Nesti et al. [38], in their paper regarding 278 adult patients with ANENs of 1–2 cm, claimed that investigations after appendectomy are not useful, there was no indication for RHC in these patients, and regional lymph node metastasis are not clinically relevant.

Indeed, no prospective study has demonstrated the effectiveness of translating indications for adults to children. Authors of the three major studies on pediatric ANENs with a total of 237, 114 and 113 patients, respectively, have suggested distinct treatment algorithms [4,11,15]. Moreover, on the basis of the cumulative information from the two reviews that included almost all the pediatric ANEN patients reported up to the year 2016 [14] and then up to the year 2022 [25], other indications were proposed for the management of this tumor.

Boxberger et al. [4] analyzed 237 German, Austrian, and Swiss children and adolescents affected by ANEN in a prospective trial (GPOH-MET) conducted from the year 1996 to 2011, and concluded that all patients with tumors greater than or equal to 1.5 cm, even if completely resected, should undergo RHC. Smaller, incompletely resected tumors should be treated with ileocecal pole resection and lymph node sampling. They identified a tumor size > 1.5 cm as the most reliable cut-off for the prediction of local lymph node metastasis, with a sensitivity of 77.8% and a specificity of 66.7%. Following this decision, there was a 50% reduction in reoperations. In their case series, no local or metastatic recurrences were observed with this approach.

Virgone et al. [15] reported on the results of 113 patients with ANENs from an Italian study named TREP between the year 2000 and 2013 in an attempt to produce comprehensive national guidelines for the management of this neoplasm. It was proposed that additional surgery should be considered for patients with an ANEN larger than 2 cm and with at least one positive result for the recommended diagnostic work-up, including urine 5-HIAA dosing, US, CT, MRI, and Octreoscan. Also, for tumors with a positive resection margin or if smaller than 2 cm but associated with pathological 5-HIAA levels and a positive Octreoscan, the authors recommended further surgery. In their study, none of the five patients with a tumor > 2 cm had positive work-up tests and only three patients with a tumor < 2 cm had a positive Octreoscan and were operated on; one of them (who also had elevated 5-HIAA) had a pericecal node of 0.9 cm with micrometastasis.

In the French multicenter retrospective study conducted by de Lambert et al. [11] on 114 children followed from the year 1988 to 2012, nineteen patients eligible for RHC were not operated on and were still alive. It was concluded that appendectomy alone, regardless of the completeness of resection, tumor site or local invasion, appeared to be sufficient for the cure of ANEN in children, without reducing life expectancy.

In addition, in the study by Sommer et al. [9], it was noted that small tumors were more often located on the tip of the appendix and that larger tumors correlated with infiltration of the outer layers of the appendix. Overall, 10% of their 40 ANEN cases underwent second surgery, none of them had residual tumors or lymph node metastasis. In accordance with previous studies, they concluded that invasion of the mesoappendix did not constitute a worsening in prognosis [3,8]. The authors suggested their own algorithm in which data regarding size and completeness of resection are associated with imaging and laboratory tests to decide if further surgery or Octreoscan are necessary before starting follow-up.

In their paper, Njere et al. [14] reviewed data of 958 patients collected from all pertinent published studies on children with ANENs up to the year 2016. Of the 189 patients that met the criteria for RHC after appendectomy, 69 had a second surgical procedure and only 43 of them underwent RHC. The other 120 patients were only observed via follow-up. There was no recurrence of tumors in either group of patients and survival was the same. The authors concluded that appendectomy without further treatment was adequate, and that tumor size, appendiceal position, lymph nodes and mesenteric involvement should not be considered in the decision to treat patients.

Van Amstel et al. [25] collected data from 29 studies after screening 667 records, with a total reported number of 1112 patients. Only 13% of cases were recognized to have high-risk ANENs, and 44% of these underwent secondary surgery. Even given the referred heterogeneity between studies, disease-free survival was 100%; thus, the authors suggested that a secondary surgery for high-risk tumors could be avoided and recommended that large and international prospective studies could confirm this assumption.

In pediatric populations, ANENs appear to display benign behavior with an excellent outcome [12,14,23,25] and, as seen above, the real importance of risk factors acknowledged for adults has been questioned in many pediatric studies. In our study, the correlation between tumor size and lymph node infiltration was confirmed, as 2.9% of patients with a tumor < 1 cm had positive nodes, compared to the 35% rate of patients with a tumor < 2 cm [14]. Nevertheless, the true clinical significance of micrometastasis in lymph nodes is still unclear, needing larger studies with long-term follow-up to determine if they represent real metastatic localizations [15]. Moreover, it has been noted that there is a lack of correlation between mesoappendix tumor invasion and lymph node involvement [3,9] and also between tumor position at the base of the appendix and metastatic diffusion [11]. Finally, the need for RHC has been questioned even in cases of tumors with vascular invasion or extension into the periappendiceal fat, due to the absence of an association with distant metastases observed by Moeretel et al. [39] and confirmed in the review by Njere et al. [14].

In our series of patients with ANENs, treatment was limited to appendectomy alone, due to the reported diameter of tumors inferior to 2 cm in all cases and the absence of high-risk tumor characteristics discovered during histology and post-surgical investigations.

Due to the limited number of prospective trials and the lack of consensus, there is wide heterogeneity in follow-up investigations and durations for children with ANENs. Moreover, follow-up information was collected in different ways between studies that reported it, i.e., by contact with general practitioners and hospital doctors or from hospital medical records [25]. As a consequence, adult methods have been considered, even though they do not appear fully applicable in children, owing to the benign behavior of ANENs in pediatric populations [23,40].

Follow-up guidelines have been described in some papers and usually consist of imaging with abdominal US, which is supported sometimes by CT scans or MRI [25]. Investigation intervals are not always reported, but they usually increase over time starting from every 3 to 6 months in the first 2 years to once a year [4,11,15,23,27,41]. 5-HIAA and CgA were also regularly tested, even if they were not considered sufficiently sensitive to detect metastasis [4,11]. Octreotide scintigraphy was usually requested if 5-HIAA or imaging was abnormal [15].

In general, no specific follow-up appears to be necessary for patients with low-risk ANENs, whereas for patients with a high-risk tumor, there is no consensus on the type of laboratory markers or imaging to be used during follow-up [25]. Authors of various pediatric studies have adopted and proposed their own follow-up recommendations.

Boxberger et al. [4] planned a follow-up that included annual abdominal US, with CgA and 5-HIAA testing, and advised the continuation of long-term follow-up to detect late-onset events.

In a French study [11], abdominal US and CT scans, Octreoscan, CGA and 5-HIAA measurements were utilized with intervals of 3 to 6 months in the first year and then once a year for 5 or 10 years. However, after evaluating their data, the authors suggested the complete avoidance of follow-up due to the lack of sensitivity in a tumor with such a low recurrence rate.

Virgone et al. [15] adopted a follow-up schema with abdominal US and 5-HIAA every 6 months for 2 years followed by annual controls for at least 10 years, leaving Octreoscan for suspicious cases.

From an analysis of previous scientific publications, it was found that the length of follow-up ranged from none to 51 years [25]. Many authors have suggested long-term follow-up regardless of the benign behavior of this tumor [4,9,15]. Prolonged monitoring is recommended due to the risk of lymph node metastasis development, even if its effectiveness in preventing tumor recurrence and improving long-term outcomes is not proven; hence, its necessity has been questioned by some authors [9,11,14].

It is important to note that only one pediatric ANEN recurrence was described in the literature [5] and the patient relapsed 18 months after second surgery with RHC and removal of a residual tumor. A third operation was performed and the presence of a well-differentiated ANEN was confirmed. The patient was alive 84 months after the last surgery.

A larger number of studies with extensive and adequate follow-up would allow us to draw evidence-based conclusions on disease-free survival and recurrence rates, and to validate guidelines for management. On the other hand, one should consider the effects of frequent and long-term investigations. One of the consequences is that repeated CT scans expose children to unnecessary irradiation and the other one is that prolonged follow-up results in a worse quality of life for both patients and parents [25,42]. In fact, for an affected child that has a 100% survival chance, after the trauma of a diagnosis of cancer, it should be of paramount importance not to negatively impact his/her quality of life with a long-term follow-up performed for the persisting fear of recurrence [43].

The duration of follow-up in our study was 10 years. Laboratory tests with 5-HIAA and CGA measurements and imaging with abdominal US and CT scans were performed in all patients at 3-month intervals for the first year and then continued (except for the one lost to follow-up) every 6 months up to the third year, and annually thereafter. Data from our series of patients with a long follow-up allowed us to confirm that well-differentiated and small ANENs have a benign course without the need for further surgery after initial appendectomy, retaining a survival rate of 100%.

## 5. Conclusions

Given the rarity of ANEN in childhood, in the literature, there are several studies with only a small number of patients and a few prospective trials relative to this tumor. As a consequence, various recommendations have been proposed by many different pediatric centers for its treatment and follow-up.

Nevertheless, based on the most recent papers, it is clear that ANENs in childhood and adolescents behave as a benign tumor, and therefore it appears that aggressive surgery after first appendectomy is not necessary, even if high-risk tumor characteristics are present. Similarly, only limited post-surgical investigations should be undertaken to better define tumors at diagnosis, and follow-ups should be reduced.

Despite the small number of patients included in our study, with this paper, we wanted to contribute to increasing the knowledge regarding pediatric patients with ANEN. In fact, it is very important to add any available data to help evaluate and validate consensus indications and to develop universally accepted protocols for the management of ANENs in this particular group of patients.

## Figures and Tables

**Table 1 cancers-16-03440-t001:** Guidelines for adult patients with appendiceal neuroendocrine neoplasms.

ENETS ^1^ Guidelines [18]	NANETS ^2^ Guidelines [7]
**RHC ^3^ if**	**RHC ^3^ if**
Tumor size > 2 cm or tumor size 1–2 cm and at least one of the following risk factors present: - positive or unclear margins - deep mesoappendiceal invasion - G2 grading - vascular invasion - lymph node metastases - tumor localized at the base	Tumor size > 2 cm or - tumor invasion at the base of the appendix - incompletely resected tumors - lymphovascular invasion - invasion of the mesoappendix - intermediate- to high-grade tumors - mixed histology (goblet cell carcinoid, adenocarcinoid) - mesenteric nodal involvement
**Appendectomy alone if**	**A** **ppendectomy alone if**
Tumor size < 2 cm and well differentiated	Tumor size < 2 cm and well differentiated
**No follow-up if**	**No follow-up if**
Tumor size < 1 cm, well differentiated and clear margins	Tumor size < 1 cm and well differentiated
**No follow-up if**	**Follow-up if**
Tumor size 1–2 cm and clear margins	Tumor size 1–2 cm and nodal metastasis or lymphovascular invasion or mesoappendiceal invasion or intermediate- or high-grade tumor or mixed histology (goblet cell carcinoid, adenocarcinoid)
**Follow-up if**	**Follow-up if**
Tumors size > 2 cm or metastasis or additional risk factors	Tumor size > 2 cm
**Follow-up intervals:**	**Follow-up intervals:**
Initially 6 and 12 months after surgery and then annually, with no duration recommended	Once between 3 and 6 months after complete resection; subsequently, patients should be assessed every 6 to 12 months for at least 7 years
**Follow-up investigations:**	**Follow-up investigations:**
CgA ^4^ or 5-HIAA ^5^ Abdominal US ^6^ to extend the intervals between CT ^7^ or MRI ^8^ scan	CgA ^4^, and 5-HIAA ^5^ CT ^7^ or MRI ^8^ recommended Octreoscan as clinically indicated

^1^ ENETS: European Neuroendocrine Tumor Society; ^2^ NANETS: North American Neuroendocrine Tumor Society; ^3^ RHC: right hemicolectomy; ^4^ CgA: chromogranin A; ^5^ 5-HIAA: 5-hydroxyindolacetic acid; ^6^ US: ultrasound; ^7^ CT: computed tomography; ^8^ MRI: magnetic resonance imaging.

**Table 2 cancers-16-03440-t002:** Characteristics of patients.

N.	Age	Sex	Histology	Tumor Size	Appendix Position	Wall Layer Invasion	Margin Positive	Presenting Symptoms	Treatment	Intra-Operative Appendix Appearance	Post-Surgery Alteration	Follow-Up Duration
1	17	F	NET ^1^, G1 ^2^	4 mm	Apex	Submucosa	No	Abdominal pain	Appendectomy	Acute phlegmonous inflammation with intraluminal abscess	Negative	10 years
2	16	M	NET ^1^, G1 ^2^	5 mm	Body	Subserosa	No	Abdominal pain	Appendectomy	Suppurative necrotic-hemorrhagic appendicitis	Transient and isolated 5-HIAA ^3^ increase	10 years
3	16	F	NET ^1^, G1 ^2^	6 mm	Apex	Subserosa	No	Abdominal pain	Appendectomy	Inflamed appendix with thickened serosa	Negative	10 years
4	11	F	NET ^1^, G1 ^2^	8 mm	Apex	Subserosa	No	Lower right abdominal pain + fever	Appendectomy	Phlegmonous appendix and abscess	Development of lymphocele	10 years
5	15	M	NET ^1^, G1 ^2^	11 mm	Body	Submucosa	No	Lower right abdominal pain + fever + vomiting	Appendectomy	Phlegmonous appendix	Negative	3 years
6	13	F	NET ^1^, G1 ^2^	10 mm	Apex	Muscularis propria	No	Lower right abdominal pain	Appendectomy	Acute appendicitis	Development of lymphocele	10 years
7	6	F	NET ^1^, G1 ^2^	5 mm	Apex	Submucosa	No	Abdominal pain + fever + vomiting	Appendectomy	Acute suppurative appendicitis	Negative	1 year (lost to FU ^4^)
8	4	M	NET ^1^, G1 ^2^	10 mm	Body	Muscularis propria	No	Lower right abdominal pain	Appendectomy	Inflamed appendix	Negative	3 years
9	11	M	NET ^1^, G1 ^2^	6 mm	Apex	Submucosa	No	Acute abdomen + fever	Appendectomy	Phlegmonous appendix	Negative	10 years
10	11	M	NET ^1^, G1 ^2^	11 mm	Apex	Muscularis propria	No	Abdominal pain + fever + vomiting	Appendectomy	Acute suppurative appendicitis with associated serositis	Negative	3 years
11	16	M	NET ^1^, G1 ^2^	3 mm	Apex	Submucosa	No	Abdominal pain + fever	Appendectomy	Inflamed appendix with thickened walls	Negative	10 years
12	15	M	NET ^1^, G1 ^2^	3 mm	Base	Muscularis propria	No	Abdominal pain	Appendectomy	Acute appendicitis	Negative	7 years
13	12	F	NET ^1^, G1 ^2^	7 mm	Apex	Muscularis propria	No	Abdominal pain	Appendectomy	Inflamed appendix	Negative	6 years
14	11	M	NET ^1^, G1 ^2^	2 mm	Apex	Muscularis propria	No	Lower abdominal pain + fever	Appendectomy	Phlegmonous acute appendicitis	Negative	3 years
15	16	F	NET ^1^, G1 ^2^	4 mm	Apex	Subserosa	No	Abdominal pain + vomiting	Appendectomy	Inflamed appendix	Negative	3 years
16	14	F	NET ^1^, G1 ^2^	6 mm	Body	Muscularis propria	No	Abdominal pain	Appendectomy	Acute appendicitis	Transient and isolated 5-HIAA ^3^ increase	2 years
17	16	M	NET ^1^, G1 ^2^	4 mm	Apex	Muscularis propria	No	Lower abdominal pain	Appendectomy	Inflamed appendix	Negative	1 year

^1^ NET: neuroendocrine tumor; ^2^ G1: grade 1; ^3^ 5-HIAA: 5-hydroxyindolacetic acid; ^4^ FU: follow-up.

## Data Availability

Data are contained within the article.

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
