# Peer review of "Characteristics and Management of Children with Appendiceal Neuroendocrine Neoplasms: A Single-Center Study"

_cancers, 2024, doi:10.3390/cancers16203440_

Round 1

Reviewer 1 Report

Comments and Suggestions for Authors

The manuscript presents an insightful exploration of ANENs in the pediatric population, based on data collected from 17 patients. This research addresses a critical gap in the understanding of these rare tumors in children, The authors are to be lauded for providing such a detailed analysis.

Some very minor comments:

- I understand that 16 out of 17 patients are alive. is the only patient who is now known to be alive the one lost to follow-up? this should be specified. If he is not, then some information on the cause of death can be provided.

- the Discussion is somehow fragmented. I suggest to have a look at its flow and reduce the number of paragraphs.

- can the Authors have a look at references published in the last 3 years, and ensure that all relevant ones are cited?

Comments on the Quality of English Language

Fine, only some proof-reading is necessary before publication

Author Response

The manuscript presents an insightful exploration of ANENs in the pediatric population, based on data collected from 17 patients. This research addresses a critical gap in the understanding of these rare tumors in children, The authors are to be lauded for providing such a detailed analysis.

Some very minor comments:

Comments 1: I understand that 16 out of 17 patients are alive. is the only patient who is now known to be alive the one lost to follow-up? this should be specified. If he is not, then some information on the cause of death can be provided.

Response 1: Thank you for pointing this out. In the abstract it is stated that” Only one patient was lost to follow-up, all other patients are alive without tumor recurrence.” (lines 35-36). In Results the relative phrase was modified to specify that one patient was lost to follow-up and the others were alive. (lines 210-2011).

Comments 2: The Discussion is somehow fragmented. I suggest to have a look at its flow and reduce the number of paragraphs.

Response 2: We agree with this comment. Therefore, paragraphs in Discussion have been reduced.

Comments 3: Can the Authors have a look at references published in the last 3 years, and ensure that all relevant ones are cited?

Response 3: We agree. Literature has been reviewed and other 3 references regarding papers published in the past 3 years were added. In detail the new references are n. 12 (cited in line 74 and 365), n.13 (cited in lines 75 and 225), and n. 24 (cited in line 214, 243, 244).

Reviewer 2 Report

Comments and Suggestions for Authors

The manuscript in case "Characteristics and Management of Children with Appendiceal Neuroendocrine Neoplasms: a Single Center Experience" dives into a single center experience with appendiceal neuroendocrine tumors that were diagnosed in 17 pediatric patients who underwent appendectomy between 1998 and 2023, with a median follow-up of 6 years. Although the novelty element is quite low and so is the number of cases analyzed, the scarce presence of this diagnosis in both adults and children justifies the publication of any additional data on this subject. The paper is well designed and written with the authors' conclusions in line with most of the available literature: this diagnosis carries a good prognosis, the tumor behaves as a benign one, appendectomy is usually the only necessary surgery. Some minor issues:

- Keywords: I suggest "appendiceal tumor" instead of "appendiceal neuroendocrine neoplasm;", it will be easier to find and it's just as rare in children

- line 252: please rephrase for better clarity

Comments on the Quality of English Language

There are several English syntax issues such as line 56 "which" instead of "this", line 14 "and" instead of "therefore", line 62 "diagnosed incidentally or after causing local symptoms..", line 64 "small tumors are considered to have a benign behaviour", line 120 "considering that for unknown reasons ..children present a better prognosis..." and others. Please have your paper read by a native English speaker

Author Response

The manuscript in case "Characteristics and Management of Children with Appendiceal Neuroendocrine Neoplasms: a Single Center Experience" dives into a single center experience with appendiceal neuroendocrine tumors that were diagnosed in 17 pediatric patients who underwent appendectomy between 1998 and 2023, with a median follow-up of 6 years. Although the novelty element is quite low and so is the number of cases analyzed, the scarce presence of this diagnosis in both adults and children justifies the publication of any additional data on this subject. The paper is well designed and written with the authors' conclusions in line with most of the available literature: this diagnosis carries a good prognosis, the tumor behaves as a benign one, appendectomy is usually the only necessary surgery.

Some minor issues:

Comments 1: Keywords: I suggest "appendiceal tumor" instead of "appendiceal neuroendocrine neoplasm;", it will be easier to find and it's just as rare in children

Response 1: The keyword was changed to “appendiceal tumor”, and the keyword “neuroendocrine tumor” was added.

Comments 2: Line 252: please rephrase for better clarity

Response 2: We agree. The sentence was modified (lines 256-258).

Comments 3: on the Quality of English Language

There are several English syntax issues such as line 56 "which" instead of "this", line 14 "and" instead of "therefore", line 62 "diagnosed incidentally or after causing local symptoms.", line 64 "small tumors are considered to have a benign behaviour", line 120 "considering that for unknown reasons children present a better prognosis..." and others. Please have your paper read by a native English speaker

Response 3: Syntax issues were addressed, those and other parts of the text were modified.

Reviewer 3 Report

Comments and Suggestions for Authors

Our colleagues propose us a paper following the diagnosis of pediatric neuroendocrine neoplasia of the appendix. We absolutely agree with the abstract and the introduction. For completeness, we remind them that it is also essential to have the value of KI-67 in the histological examination. It should also be noted that the neuroendocrine tumor may not be evident for two essential reasons, the first that it may not be productive, the second that the hormone produced may not be stoichiometrically perfect so it does not adhere to the target receptors, furthermore those under 2 cm in diameter are often painless, this explains the total absence of symptoms. Furthermore, in the follow-up, be careful to use octreotide because if the cells do not have receptors, perhaps because of grading 3, there will be no response. Gallium scintigraphy is better as it is more reliable. Finally, I would add some reference to possible medical therapy, should it happen that it needs to be put into practice. Somatostatin with analogues recommended? Drugs that are absolutely manageable but that expose patients to possible complications (PMID: 38051513 to be cited in the bibliography), the less manageable everolimus widely used in adulthood for these diseases (PMID: 29757017 to be cited in the bibliography but the side effects. Other protocols?

The paper does not need many corrections, since the authors propose an attitude to be taken and that we can share in the absence of guidelines, we ask them to spend a few more words on the therapeutic treatment, if some surgeon happens to find himself faced with problems of advanced disease given that, the practice of treating appendicitis, once defined as catarrhal or in any case moderately inflamed, with only antibiotics is increasingly spreading and we could find ourselves in some subsequent surprises. the weak point is constituted by the number of patients. Excellent iconography, good English, the bibliography is a solid basis for the entire paper.

Author Response

Our colleagues propose us a paper following the diagnosis of pediatric neuroendocrine neoplasia of the appendix. We absolutely agree with the abstract and the introduction.

Comments 1: For completeness, we remind them that it is also essential to have the value of KI-67 in the histological examination.

Response 1: Thank you for pointing this out. Ki67 index values of the 10 patients analyzed are now reported in line 185.

Comments 2: It should also be noted that the neuroendocrine tumor may not be evident for two essential reasons, the first that it may not be productive, the second that the hormone produced may not be stoichiometrically perfect so it does not adhere to the target receptors, furthermore those under 2 cm in diameter are often painless, this explains the total absence of symptoms.

Response 2: The above concept in now described in lines 61-62.

Comments 3: Furthermore, in the follow-up, be careful to use octreotide because if the cells do not have receptors, perhaps because of grading 3, there will be no response. Gallium scintigraphy is better as it is more reliable.

Response 3: The use of Gallium scan has been addressed in lines 112-114, and reference n. 22 was added.

Comments 4: Finally, I would add some reference to possible medical therapy, should it happen that it needs to be put into practice. Somatostatin with analogues recommended? Drugs that are absolutely manageable but that expose patients to possible complications (PMID: 38051513 to be cited in the bibliography),the less manageable everolimus widely used in adulthood for these diseases (PMID: 29757017 to be cited in the bibliography but the side effects. Other protocols? The paper does not need many corrections, since the authors propose an attitude to be taken and that we can share in the absence of guidelines, we ask them to spend a few more words on the therapeutic treatment, if some surgeon happens to find himself faced with problems of advanced disease given that, the practice of treating appendicitis, once defined as catarrhal or in any case moderately inflamed, with only antibiotics is increasingly spreading and we could find ourselves in some subsequent surprises.

Response 4: Medical treatment for advanced disease issue is now dealt with in lines 225-228 and 3 references were added (n. 31,32, and 33).

the weak point is constituted by the number of patients. Excellent iconography, good English, the bibliography is a solid basis for the entire paper.